# Colloidal CdSe nanocrystals are inherently defective

Tamar Goldzak[1,2,3], Alexandra R. McIsaac [1,3] & Troy Van Voorhis [1✉]

Colloidal CdSe nanocrystals (NCs) have shown promise in applications ranging from LED displays to medical imaging. Their unique photophysics depend sensitively on the presence or absence of surface defects. Using simulations, we show that CdSe NCs are inherently defective; even for stoichiometric NCs with perfect ligand passivation and no vacancies or defects, we still observe that the low energy spectrum is dominated by dark, surface-associated excitations, which are more numerous in larger NCs. Surface structure analysis shows that the majority of these states involve holes that are localized on two-coordinate Se atoms. As chalcogenide atoms are not passivated by any Lewis base ligand, varying the ligand should not dramatically change the number of dark states, which we confirm by simulating three passivation schemes. Our results have significant implications for understanding CdSe NC photophysics, and suggest that photochemistry and short-range photoinduced charge transfer should be much more facile than previously anticipated.

[1] Department of Chemistry, Massachusetts Institute of Technology, Cambridge, MA, USA. [2] Present address: Department of Chemistry, Columbia University, New York, NY, USA. [3] These authors contributed equally: Tamar Goldzak, Alexandra R. McIsaac. ✉email: tvan@mit.edu

Colloidal semiconductor nanocrystals (NCs), or quantum dots, are solution-processed materials that typically consist of hundreds of semiconductor core atoms surrounded by a ligand shell that both passivates the surface and imparts solubility. Semiconductor NCs exhibit discrete optical spectra that are qualitatively different from the bulk due to quantum confinement[1–3]. The NCs' spectrum and optical properties can be tuned by changing the NC size, shape, and composition[4–6]. Due to their tunable spectrum and high luminescence, colloidal NCs have been successfully used in many applications and optoelectronic devices[7], such as solar cells[8–11], photodetectors[12,13], light-emitting diodes[14,15], displays[16], and biological sensing and imaging[17,18].

The low-energy absorption spectra of NCs are typically dominated by a bright peak or peaks[3,19] and the way in which these peaks shift with NC size, as well as the spacing between them, can be qualitatively explained by simple particle-in-a-sphere (PiS) models[20]. For this reason, NCs are sometimes referred to as artificial atoms[2], despite the fact that, aside from their absorption spectra, NC photophysics is very different from atomic photophysics: the photoluminescence (PL) spectra of NCs often display a significant Stokes shift[21]; the PL quantum yield (QY) is significantly less than unity[22]; and under constant illumination, single NC PL displays an on-off intermittency known as blinking[23–26].

In each of these situations, the unusual photophysics of NCs is intimately tied to the existence of surface defect states that compete with the bulk-like PiS states. For example, blinking is thought to be tied to surface charge defects[25,27], while PL QY can be improved by using ligands that better passivate the surface[28]. The chemical nature of these defects has been the source of much study and speculation: incomplete ligand passivation[29], off-stoichiometry NCs[30], charging[25,27], vacancy formation[31,32], and dopants[33] have all been implicated as potential sources of surface defect states. The common assumption is that defects arise from imperfections in the synthesis—if one could only create dopant-free NCs with perfect passivation, these surface states could be removed. Indeed, this picture is supported by the fact that core-shell NCs have dramatically improved PL QY and significantly reduced blinking[22,34].

It has been shown computationally that unpassivated surface Cd atoms create electron traps and unpassivated surface Se atoms create hole traps in the ground state band structure[35–40]. These traps will affect electron transport, although often the surface of a NC will rearrange or undergo self-healing to eliminate mid-gap traps, resulting in a clean band gap[29,38]. Despite many experimental and theoretical works suggesting that surface traps strongly affect the optical properties of NCs, these effects have been largely unexplored from an ab initio computational point of view. A few studies have used ab initio tools to simulate optical absorption spectra of CdSe quantum dots[41–43] but the connection of the computed spectra to surface states or the processes described above has yet to be addressed.

Here, we present computational evidence that CdSe NCs are inherently defective. That is, we present data suggesting that CdSe nanoparticles possess dark surface excitations that cannot be eliminated by any commonly used ligand for surface passivation and are not associated with any compositional defect. Using time-dependent density functional theory (TD-DFT)[44], we simulate several stoichiometric, uncharged, fully passivated, fully relaxed NCs of various sizes and find that despite clean band structures with no mid-gap states, in all cases, most of the low lying excited states are surface associated and optically dark. By examining the size dependence of our results, we conclude that these states become more prevalent as the size of the NC increases. Careful examination of the NC geometry reveals that, despite significant

surface rearrangement, there remain several undercoordinated surface Se atoms. Examination of the electron and hole spatial distributions reveals that the majority of the dark states have holes that are strongly localized on these undercoordinated surface Se atoms. This observation explains why these defects are essentially unavoidable in most core-only CdSe NCs—no commonly used NC ligand coordinates with the chalcogenide. Indeed, simulating three different passivation schemes, we find that these defect states are present in every case. Our results have significant implications for the understanding of semiconductor NC photophysics and could have a significant impact on the design of NC-based photcatalytic and photon upconversion systems.

## Results

We created stoichiometric, defect-free CdSe NCs of various sizes following a protocol used previously for PbS[45] and CdSe NCs[46]. Briefly, we use the Wulff construction procedure to create a quantum dot of a given diameter cut from the bulk Wurtzite CdSe structure. We remove any singly coordinated atoms from the structure, and check to make sure it is roughly spherical and is stoichiometric. We then passivate all of the surface Cd's with ligands, and fully relax the structure using density functional theory (DFT). Careful analysis of the resulting atomic structures and band structures (see Supplementary Note 2) confirms that this process results in NCs that do not have obvious defects such as charging or detached ligands, and have a clean band gap with no mid-gap trap states for electrons or holes.

For a diameter of 2 nm, this procedure results in a $Cd_{91}Se_{91}$ NC, whose TD-PBE0[47] excitation energies with their corresponding oscillator strengths and absorption spectrum are shown in Fig. 1 (black sticks and solid red line, respectively). More information on calculating the absorption spectrum is presented in the Methods section. The absorption peak around ~2.95 eV is in good agreement the experimental peak for NCs of comparable size[19], once one accounts for the fact that PBE0 slightly overestimates the band gap of CdSe[48]. To the red of the bright peak, there is a weakly absorptive tail that is typically not observed in experiment. However, due to the fact that these states are quite dim and broad, it would be difficult to differentiate these from background noise in any realistic situation. Thus, it seems likely that, in addition to the bright state, the simulations are revealing a set of states that would simply be overlooked in the experimental absorption spectrum. In addition to the absorption spectrum, which only reveals optically active states, Fig. 1 displays the density of transitions—that is, the density of excited states at a given energy—as a broadened curve (blue line). Clearly there are many dark states both at low energies —below the lowest bright state—as well as at energies comparable to and even above the first bright state. Absorption spectra of this type have been computed before for $Cd_{33}Se_{33}$ NCs and dark states below the bright state were also observed[41,42], but the nature of these non-bright states has been largely unexplored.

In order to characterize these states, we have examined the attach and detach densities[49] for the transitions in Fig. 1. Roughly speaking, the detach density is the density of the excited hole and the attach density is the density of the excited electron. These excited state densities can be broken down into charge contributions from different atoms, in a similar manner to Löwdin charge analysis for the ground state[50]. We can then use the inverse participation ratio (IPR) to estimate the number of atoms that contribute significantly to the electron or hole density:

$$IPR^e \equiv \frac{1}{N\sum_A (q_A^e)^2} \quad IPR^h \equiv \frac{1}{N\sum_A (q_A^h)^2} \quad (1)$$

where $q_A^e$ is the charge from the electron (attach) density associated with atom $A$, $q_A^h$ is the charge from the hole (detach)

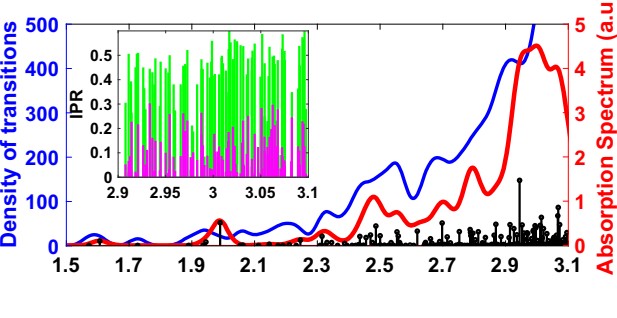

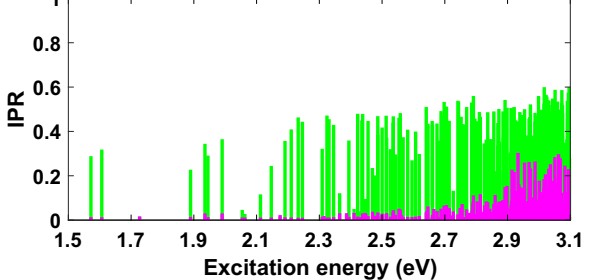

**Fig. 1 Spectrum and IPR for the 2 nm, methylamine-passivated Cd$_{91}$Se$_{91}$.** Top: Absorption spectrum in arbitrary units (red) and density of transitions (blue). Black sticks show individual excited states, whose oscillator strengths have been broadened to create the absorption spectrum. Note the high density of states with little or no oscillator strength (i.e. dark states). Bottom: Inverse participation ratio (IPR) for electrons (green) and holes (magenta) for each excitation. A larger IPR corresponds to a more delocalized wavefunction. Clearly the dark, low-energy states are associated with localized states, while the bright states are significantly delocalized. Inset: Enlarged plot of the IPR for the higher-energy part of the excitation spectrum, i.e. $E_{ex} > 2.9$ eV.

density associated with atom $A$, and $N$ is the total number of semiconductor atoms (e.g. 182 for the 2 nm NC). If the electron or hole is delocalized equally over all atoms then the IPR will be equal to 1, since $q_A = \frac{1}{N}$ for all atoms. Meanwhile, if the electron or hole is localized on a single atom, IPR $= \frac{1}{N}$. Thus, the IPR gives a single number that allows us to distinguish surface (localized, low IPR) from bulk (delocalized, high IPR) states[40], as well as recognizing when something is between these two limits.

The electron and hole IPRs for the 2 nm NC are shown in the bottom panel of Fig. 1 (green and magenta, respectively). We directly visualize the electron and hole densities for a number of representative states in Fig. 2. Qualitatively, these hole-to-electron excitations can be divided into three types of states: surface-to-surface (Fig. 2a), surface-to-bulk (Fig. 2b, d), and bulk-to-bulk (Fig. 2c), such that a surface or bulk state corresponds to a density localized on the surface of the NC or delocalized over the whole NC, respectively. Across the entire energy range of the spectrum, the majority of the states we observe for this 2 nm NC are surface-to-bulk states. Quantitatively, the IPR clearly shows that the hole density is localized for almost all excitations below ~2.95 eV, and Fig. 2a, b show that these densities are localized on the surface of the NC. Above 2.95 eV, the IPR shows that some of holes are now quite delocalized, with 25% of the atoms routinely being involved in each excitation. Note that even the delocalized holes are almost entirely confined to the Se atoms, so 25% of the total number of atoms corresponds to 50% of the practically available sites, meaning that these correspond to quite delocalized hole states, as shown in Fig. 2c. We will use 25% as a cutoff to distinguish between localized and delocalized states. However, even at high energies there are a large number of localized hole states interspersed with the delocalized states, as well as states that are

hybrids between localized and delocalized states, which can be seen in Fig. 2d. The electron density is localized in a few cases (Fig. 2a), but is typically very delocalized even for low-energy states, and remains delocalized for the vast majority of bright states (Fig. 2b–d).

Based on this analysis, the low-energy excitations can all be classified as either surface-to-surface or surface-to-bulk. This result is consistent with the orbital density of states (Supplementary Fig. 3)—the lowest conduction band orbitals are all localized on the surface for this NC. However, it is important to note that these surface orbitals are not in the gap; rather, they form the edge of the quasi-continuous band of available energies in these NCs. Furthermore, the TDDFT excited electron and hole states are comprised of a linear combination of orbitals, and thus a few surface orbitals lead to a large number of surface hole states. This analysis thus reveals a situation in which the dark states are primarily associated with highly localized hole densities. We observe that, among the dark states, those with higher oscillator strength tend to be surface-to-bulk excitations (cf. the small peak at 2 eV, (see Fig. 2b)), which is consistent with our physical intuition that the electron and hole wavefunctions must overlap in space to produce significant absorption intensity. Above 2.95 eV, we observe the emergence of bulk-to-bulk states that qualitatively resemble the wavefunctions one would predict based on the PiS model. This observation explains why the PiS model is so successful in explaining the absorption spectra of NCs even when most of the states are not PiS-like: as long as the bright states are PiS-like, changes in the absorption spectrum will be largely captured by ignoring all the non-PiS states. However, as should be clear from the discussion above, while the spectrum is largely controlled by bulk-to-bulk PiS states, the low-energy transitions are dominated by surface states. Furthermore, the brightest excited state—the 143$^{rd}$ excited state—is half way between a localized surface state and a delocalized PiS state.

Our NCs are smaller than the smallest CdSe NCs typically used in devices (~3 nm), so one might question the relevance of our findings to realistic NCs. We have therefore examined the size dependence of our results. For computational reasons, we cannot simulate NCs that are significantly larger than Cd$_{91}$Se$_{91}$ and so instead we looked at smaller NCs to explore this trend. Figure 3 shows the absorption spectra for Cd$_{33}$Se$_{33}$ (1.3 nm diameter) and Cd$_{38}$Se$_{38}$ (1.4 nm diameter) NCs. In order to facilitate comparison with the Cd$_{91}$Se$_{91}$ NC from Fig. 1, we chose the same number of excitations (260) in each case, meaning that progressively higher energy states are included for progressively smaller NCs. The first immediate observation is that the number of low-energy surface-associated states below the first bright bulk state is clearly increasing with NC radius. For the smallest NC, below the first bright bulk state at ~3.2 eV there are 21 surface holes (78% of the hole states in this regime) with IPR < 0.25; for the medium-sized NC there are 42 surface hole states (93%) below the first bright bulk excitation at ~3.1 eV; for the largest NC, there are 142 (99%) surface hole states below the bright excitation at ~2.95 eV, as well as a number of surface electron states that were not present in the smaller NCs. This trend also extends to the higher energy part of the spectrum; 16% of the excitations above the first bright excitation have holes on the surface in the smallest NC, 33% have holes on the surface for the medium-sized NC, and 92% for the larger NC. This observation suggests that for compositionally perfect NCs in the experimental size range, localized surface-like states are likely to be even more prevalent than our results here would suggest.

We have been careful to create the most perfect core-only NC that could be experimentally realized—stoichiometric, uncharged, and fully passivated with Lewis base ligands—yet we still see surface states. In order to understand this fact, we analyzed the

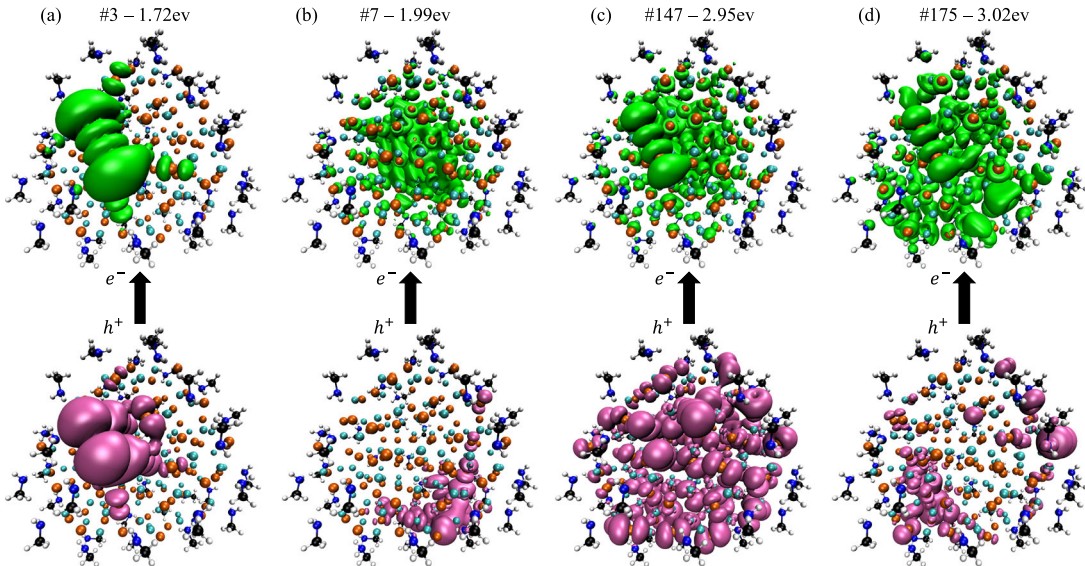

**Fig. 2 Representative attach and detach densities for Cd$_{91}$Se$_{91}$.** Attach (top-green) and detach (bottom-magenta) densities for four representative excited states of the 2 nm diameter methylamine-passivated Cd$_{91}$Se$_{91}$ NC, showing the locations of the electron and hole, respectively. **a** 3$^{rd}$ excited state. **b** 7$^{th}$ excited state. **c** 147$^{th}$ excited state. **d** 175$^{th}$ excited state. These correspond to surface-to-surface, surface-to-bulk, bulk-to-bulk, and surface-to-bulk states, respectively. The bright states are delocalized in a similar way to the 147$^{th}$ excitation, in the way that would be expected from the particle-in-a-sphere model. The colors of the atoms in the NC are: Cd-cyan, Se-orange, C-black, N-blue, H-white.

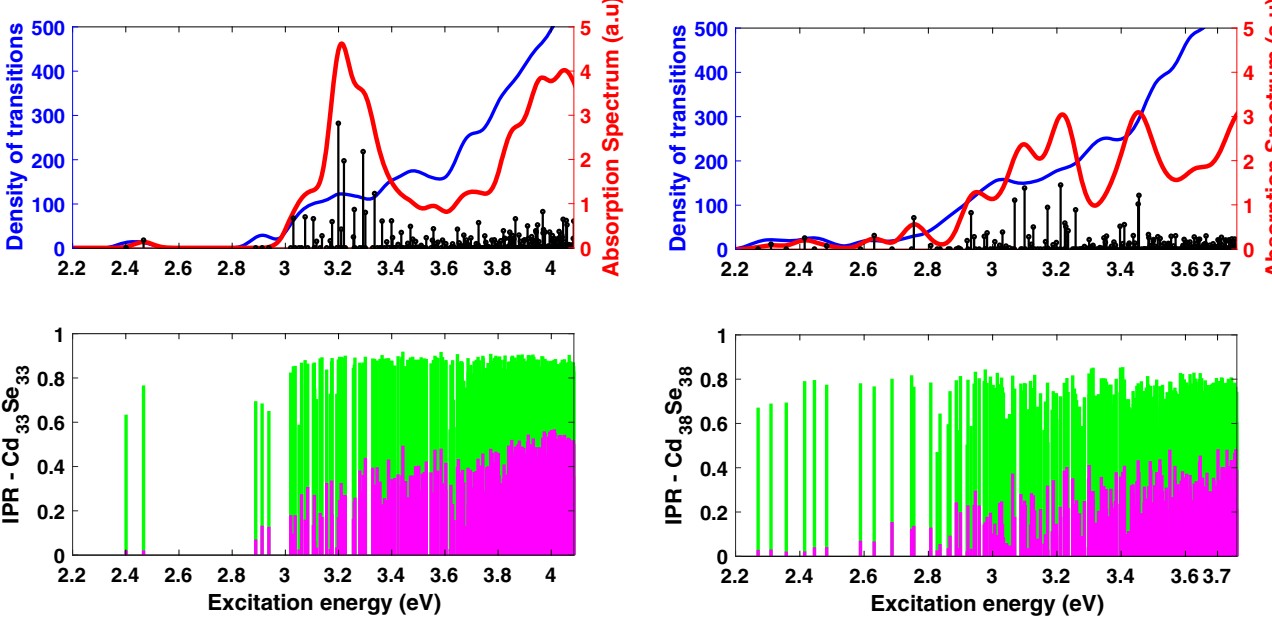

**Fig. 3 Effect of dot size on the spectrum and IPR.** Top: Absorption spectrum in arbitrary units (red), density of transitions (blue), and individual excitations (black sticks) for Cd$_{33}$Se$_{33}$ (left) and Cd$_{38}$Se$_{38}$ (right) with methylamine ligands. Bottom: Inverse participation ratio (IPR) for Cd$_{33}$Se$_{33}$ (left) and Cd$_{38}$Se$_{38}$ (right) with methylamine ligands. In each case, the number of excited states is chosen to be equal to that of the 2 nm NC in Fig. 2. The number of low energy, surface-associated transitions clearly increases with NC size.

bonding pattern in our NCs. The one obvious defect in any NC is that some of the atoms at the surface will be undercoordinated; Cd and Se atoms will all be four-coordinate in the bulk, but at the surface this necessarily cannot be the case. In particular, surface atoms with less than three neighbors have been shown to routinely create traps in the ground state band structure of NCs[35–40]. Our strategy of passivating all surface Cd atoms and relaxing the geometry (which has been shown to eliminate undercoordinated Se atoms) is aimed at eliminating these band structure defects. In

agreement with previous findings[40,41,51], we can confirm that this approach results in a band structure free of mid-gap states (see Supplementary Fig. 3). However, for every NC we have studied there are two-coordinate Se atoms at the surface. They do not dominate the ground state band structure, but they do dominate the excited states. In Fig. 4, we plot the fraction of the Löwdin charge on the two-coordinate Se atoms for all of the hole densities in the Cd$_{91}$Se$_{91}$ spectrum. We find that the low-energy surface states tend to have hole densities that are strongly localized on the

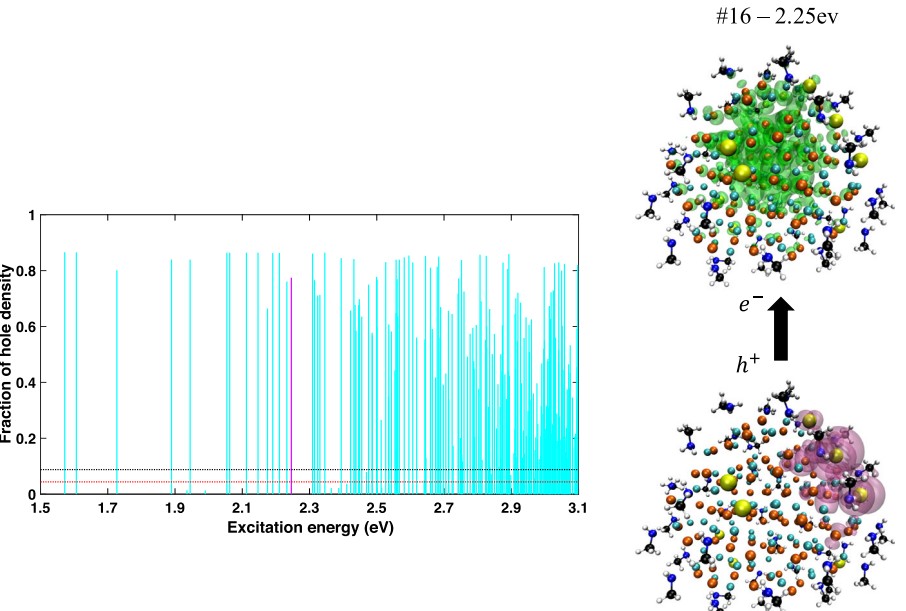

**Fig. 4 Effect of undercoordinated Se atoms on the excited states of Cd$_{91}$Se$_{91}$.** Left: Fraction of hole charge density on the eight Se atoms that are two-coordinate in the Cd$_{91}$Se$_{91}$ NC with methylamine ligands. The red and black dotted lines illustrate the expected fraction based on the assumptions that the hole is distributed equally among all Se or all Cd and Se atoms, respectively. Clearly, most of the low-energy states preferentially localize on the two-coordinate Se atoms. Right: Attach (green) and detach (magenta) densities of the 16$^{th}$ excited state (marked with a magenta line in the left hand figure). The undercoordinated Se atoms are highlighted in yellow. Clearly the hole is specifically localized near a subset of these atoms.

undercoordinated Se atoms. This is illustrated qualitatively in the right panel of Fig. 4, which shows the electron and hole densities for the 16th excitation; the hole density is clearly localized on the undercoordinated Se atoms, which are highlighted in yellow. Clearly, the majority of the low-energy states in these NCs can be attributed to the presence of these two-coordinate Se atoms. Interestingly, we also find that there are some electron densities that localize on these undercoordinated Se atoms (Supplementary Fig. 2), and so they can also serve as electron traps (see Fig. 2a). But this effect is less dominant than the localization of the hole densities on the undercoordinated Se.

Now, given that no commonly used solubilizing ligand passivates the chalcogenide atoms in NCs, one would therefore assume that no ligand would be effective at removing these low-energy states. That is, one would assume that these dark states are not a feature of methylamine ligands in particular, but rather would be present for CdSe NCs with any organic passivating ligand on the Cd atoms. To test this, we repeated our simulations for NCs passivated with trimethyl phosphine oxide (Me$_3$PO), intended to mimic the commonly used TOPO ligand, as well as completely unpassivated NCs. For computational expediency, we used the medium-sized Cd$_{38}$Se$_{38}$ NCs for this comparison, and the results are shown in Fig. 5. Clearly both the Me$_3$PO-passivated NC and the bare NC qualitatively resemble the amine-passivated counterpart in Fig. 3—there are a large number of dark transitions below the bright transition (39 surface holes for Me$_3$PO and 53 for the bare NC). Indeed, it is clear that methylamine is actually the best ligand (amongst those we have studied) for suppressing these dark states. This observation is consistent with the experimental observation that amine-passivated NCs have among the highest QYs for stoichiometric core-only structures[52,53]. Further, by looking at the IPRs of the electron and hole densities, we find that, as before, the dark states are all surface-associated in some fashion. We thus conclude that, for all practical purposes, stoichiometric core-only CdSe NCs are intrinsically defective—there is no Lewis base ligand that removes the surface states even when the passivation is complete.

## Discussion

The existence of a dense manifold of dark states has the potential to dramatically change our understanding of processes like multiple exciton generation[8] (where the presumed lack of intermediate states is assumed to stabilize the biexciton state) and blinking[24–26] (where the dark state is typically assumed to be a single state rather than a quasi-continuum). At the same time, the fact that these states exist at the surface will have an outsized influence on photochemistry initiated by the NCs[54,55] as well as processes like upconversion[56], energy transfer[23], and electron transfer[57,58], where proximity to the surface drives function.

The presence of such a large number of dark surface states is also consistent with a number of experimental observations. First, there is the relatively large (up to 0.1 eV) nonresonant Stokes shift observed in most as-synthesized pristine CdSe NCs that appears on the picosecond timescale[59,60]. It has been clearly argued that the energy difference between the absorbing and emitting states is due to the exciton fine structure[20,59,61]. However, in order to facilitate an ultrafast 0.1 eV relaxation in the absence of large atomic distortions, one would typically expect there to be various intermediate (I) electronic states between the absorbing (A) and emitting (E) states, so that the relaxation involves a series of small steps (A → I$_1$ → I$_2$ → . . . → E), rather than a single large one. Such a series of states is not easily accommodated in the PiS or artificial atom picture of NCs, in which only a handful of well-separated electronic states exist and the states get further apart as one approaches the lowest excitation. However, in a picture of NCs that contains both bright bulk-to-bulk states and dark surface-associated states, the ubiquitous Stokes shift of colloidal NCs is easily explained. There are many surface states of varying brightness below the strongly absorbing bulk-to-bulk transitions. These states can play the role of the intermediate states in the mechanism above. Thus, our findings provide an easy explanation for the observed Stokes shift dynamics.

Another important role of dark states is in explaining the relatively low QY and blinking behavior of CdSe NCs. Again, in the PiS or artificial atom models, these effects are hard to explain,

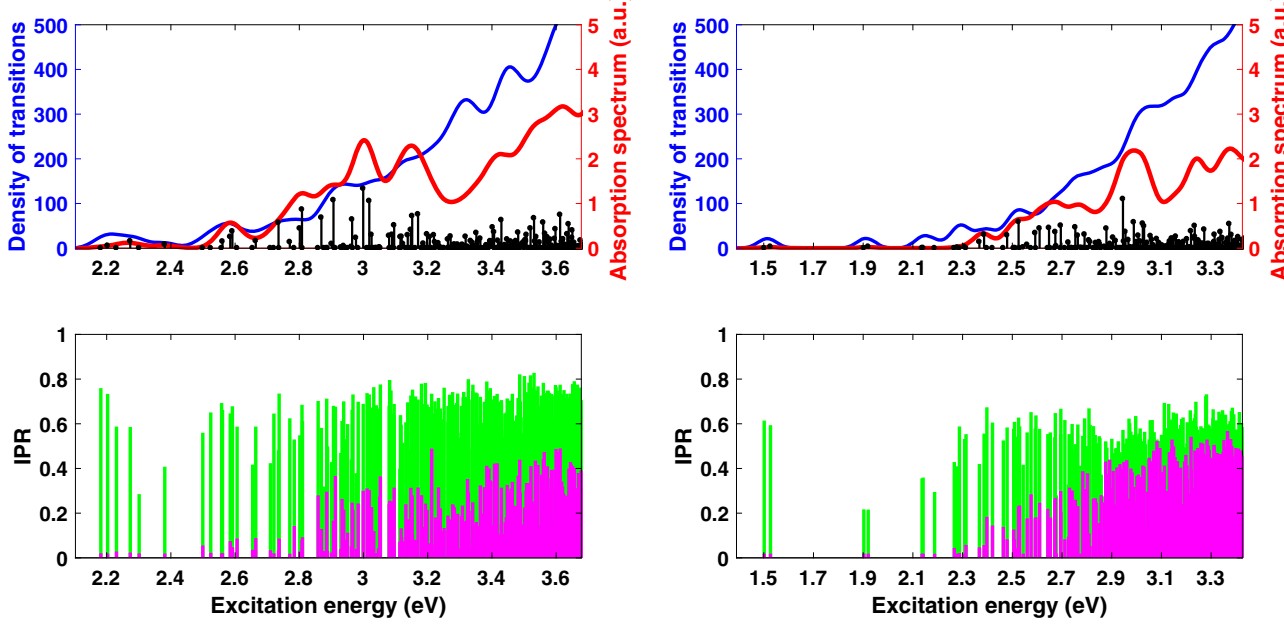

**Fig. 5 Effect of ligands on the spectrum and IPR.** Top: Absorption spectrum in arbitrary units (red), density of transitions (blue), and individual excitations (black sticks) for $Cd_{38}Se_{38}$ with $Me_3PO$ ligands (left) and $Cd_{38}Se_{38}$ with no ligands (right). Bottom: Inverse participation ratio (IPR) for $Cd_{38}Se_{38}$ with $Me_3PO$ ligands (left) and $Cd_{38}Se_{38}$ with no ligands (right). In each case, the number of excited states is chosen to be equal to that of the 2 nm NC in Fig. 2. The number of low energy, surface-associated transitions is similar to that of the $MeNH_2$ case shown in Fig. 3.

as the lowest state is typically bright or at the very least well separated from any dark states it might transition to. Our findings suggest the spectrum of low-lying states in colloidal CdSe NCs is much more complex than the simple PiS model predicts—with even the most perfect NCs having a fairly dense manifold of low-lying states with very long emission lifetimes. Those long lifetimes give a larger window for non-radiative processes to interfere with emission. Relaxation to the ground state would reduce the QY, while photoreduction could turn the emission off. Thus the presence of dark surface states also provides a facile explanation of blinking and QY in CdSe NCs.

Among core-only NC structures, it has been shown experimentally that NCs with a Cd-rich surface provide the highest QY for CdS[30] and CdTe[62] NCs—probably because the additional Cd atoms passivate some 2-coordinate chalcogenide atoms[40,51]. Indeed, those results lead to a somewhat different picture of a perfect NC—one in which the entire surface is passivated with $CdX_2$ (X = Cl, I) ligands[40,51] in what amounts to a $CdSe/CdX_2$ core-shell structure. This is also consistent with our result that there are many dark surface states in what is traditionally considered a perfect quantum dot, which are caused by underpassivated Se atoms.

The presence of a large number of dark surface states is consistent with previous theoretical calculations on CdSe. Previous ground state studies have noted that, despite a clean-looking band gap, the HOMO was localized on the NC surface[38], while for less perfect dots, it has been noted that the leading cause of mid-gap traps is the presence of two-coordinate Se atoms[30,40]. We will emphasize that in these studies, defects were intentionally introduced in such a way as to create deep traps in the band gap. In our case, surface defects have been avoided insofar as geometrically possible (to make a "perfect" NC) and the resulting band structure does not show the same trap states. Thus, while our NCs do have surface defects in the form of two-coordinate Se atoms, in the language of previous studies they are defect-free. That is, in agreement with previous studies, we find that trap states can be eliminated from the band structure of CdSe NCs. Looking at the

small number of computational studies of the excited states of these NCs, TDDFT studies of $Cd_{33}Se_{33}$ with methylamine ligands had also found a long tail in the absorption spectrum[41,42], and studies using TDLDA for NCs with no ligands found an extremely broad and long tail[43], though these works did not explore the nature of these states. Further, when one uses formate anions to passivate Cd and $H^+$ to passivate Se, the TDDFT spectrum becomes much cleaner than when methylamine alone is used[42]. While this geometry is likely not stable in practice, the reduction in dark states that comes from artificially passivating the Se sites is consistent with our observation that undercoordinated Se atoms contribute heavily to the optically dark states in these NCs.

Finally, our finding that the number of low-lying surface states increases with NC size runs counter to the experimental result, where smaller NCs are typically more defective than larger NCs[32]. This difference is easily understood when one remembers that our simulations are always dealing with perfect NCs—fully passivated, stoichiometric and non-defective structures. It is fairly well accepted that below a certain size, it is difficult to control the number of defects in CdSe NCs[63]. Thus, as the NCs get smaller, our simulated systems become less and less similar to the experimental system. These differences explain the divergence between the two results. Fortunately, we can extrapolate our results to the size range (3–20 nm) in which the experimental NCs are thought to be most nearly perfect. Doing this, we find that surface states are even more prevalent for those large NCs than the results here would suggest.

Taken together, our results suggest that the PiS/artificial atom model of colloidal NCs is incomplete. A qualitatively correct picture will need to accommodate both the bright bulk-to-bulk PiS-like states as well as a dense manifold of surface-associated states. At present there are no simple models that can accomplish this, although recent results using random matrix theory[64] are potentially one good first step.

In conclusion, we present computational evidence that even perfectly passivated CdSe NCs possess inherent surface defects. By simulating a large ($Cd_{91}Se_{91}$) NC fully passivated with amine

ligands using hybrid TDDFT, we find that the overwhelming majority of the low-lying states are associated with the surface—the brightest transition is the 143$^{rd}$ excited state of the NC. By comparing to the results for smaller (Cd$_{33}$Se$_{33}$ and Cd$_{38}$Se$_{38}$) NCs we observe that, as long as all the NCs are similarly passivated, the number of low-lying surface states tends to increase with the size of the NC. Careful correlation with the chemical structure reveals that the majority of these defects are associated with hole densities localized on a handful of two-coordinate Se atoms. As no common ligand passivates the surface Se atoms, we conclude that the majority of these defects are therefore inherent to core-only CdSe NCs. We test this by performing calculations for phosphine oxide passivated NCs and bare NCs, and confirm qualitatively similar results in each case.

The implications of these results are wide-ranging. The surface availability of electrons and holes in photoexcited CdSe NCs is likely to be much higher than previously anticipated, as even a NC that ultimately emits from a bulk-like state has likely passed through numerous surface localized states to get there. The stability and coherence of bi- and tri-exciton states are likely to be much lower than expected as they couple to a much denser manifold of excitations. The QY of core-only CdSe NCs is likely intrinsically limited by these optical traps. Future work aimed at elaborating and quantifying these effects will be highly influential.

Leading up to that aim, there are several aspects of the present study that can be expanded upon. First and foremost, though the Cd$_{91}$Se$_{91}$ NC is quite large by simulation standards, it still falls short of the experimentally relevant size range. Thus, simulations on somewhat larger NCs would help shed more light on the way that these effects reveal themselves in real NCs. Second, here we have only explored equilibrium structures in an attempt to characterize the behavior for completely relaxed NCs. It would be interesting to examine the impact that structural distortions—including simple thermal fluctuations—have on the electronic states.

Finally, we have only looked at a handful of passivation schemes in this work and it would be instructive to explore others. In particular, it has been shown that core-shell CdSe quantum dots exhibit greatly enhanced QY and suppressed blinking, which has been attributed to the shell's ability to passivate both Cd and Se atoms[22]. Here we have shown that underpassivated Se atoms are responsible for the majority of the hole traps in the Cd$_{91}$Se$_{91}$ dot, but it would be interesting to explore whether this is also true for other sizes and ligands. It would also be extremely valuable to perform analogous simulations for CdS/CdSe and ZnS/CdSe NCs to see what defects, if any, persist in pristine core-shell NCs. Another obvious extension would involve exploring other semiconducting NCs to see how universal this phenomenon is. Recent calculations on PbS, for example, suggest that low-lying surface states may be present in those NCs as well[51,56] whereas it has been known for many years that hydrogen terminated Si NCs are defect free[65]. Similar calculations on CdS, GaAs, InAs and other compositions could help elucidate which properties of the material tend to lead to inherent defects. Does faceting help or hinder surface passivation? Are binary semiconductors inherently more defective? Is it simply necessary to passivate every single surface atom to remove defects?

These types of chemical and physical questions will then guide the design of simple models that are capable of incorporating the effects of surface states on the photophysics of semiconductor NCs. Such models will play a key role in understanding and designing new devices based on these fascinating materials.

## Methods

**Creating the NC structures**. All NC starting structures, except for the Cd$_{33}$Se$_{33}$ NC, were carved from bulk CdSe with the wurtzite structure. The Cd$_{38}$Se$_{38}$ NC has a diameter of about 1.4 nm and the Cd$_{91}$Se$_{91}$ NC has a diameter of about 2 nm. For the passivated NCs, we passivate every surface cadmium with ligands, in accordance with other simulations and experimental results[29]. The Cd$_{33}$Se$_{33}$ NC starting structure was obtained from the supplementary information of ref. [29], but ligands were replaced due to optimization convergence issues.

**DFT methodology**. All calculations are performed using the QChem software package[66].

*Geometry optimizations*. Geometry optimizations were performed at both the PBE[67]/LANL2DZ[68–71] and PBE[47]/LANL2DZ levels of theory. All NCs were optimized until converged by the default QChem geometry optimization thresholds.

*TDDFT*. All TDDFT calculations were performed with the PBE0 functional and the LANL2DZ basis set and effective core potential. PBE0 was chosen as it is a hybrid functional that incorporates exact exchange, which is important for effective treatment of the electron–hole interaction. Validation studies were conducted to ensure that this level of theory performed well for this problem (see Supplementary Notes 3 and 4). For the Cd$_{91}$Se$_{91}$ NC, some calculations used a reduced single excitation space (see Supplementary Note 6) to overcome a memory limitation.

**Density of transitions and absorption spectra**. To calculate the density of transitions, we apply Gaussian broadening to the calculated excitation energies ($E_{ex}$), and sum the Gaussian distributions:

$$\text{DOT} = \sum_{ex} \frac{1}{\sqrt{2\pi\sigma^2}} \exp\left(-\frac{(E - E_{ex})^2}{2\sigma^2}\right) \quad (2)$$

To calculate the absorption spectrum, we do the same, but weight the transitions by their oscillator strength ($\omega$):

$$\text{Spectrum} = \sum_{ex} \frac{\omega}{\sqrt{2\pi\sigma^2}} \exp\left(-\frac{(E - E_{ex})^2}{2\sigma^2}\right) \quad (3)$$

For both calculations, we choose $\sigma$ to be 35 meV, on the order of the experimental broadening[29]. For the associated stick spectra, the oscillator strengths have been multiplied by 15 so as to be visible on the same axes as the absorption spectra.

**IPR**. To calculate the IPR, we first conduct a Löwdin charge analysis on the attach and detach densities for all excited states, as implemented in QChem[50]. For each excitation, analysis of the attach density gives atomic charges related to the excited electron ($q_A^e$ for atom A), and analysis of the detach density gives atomic charges related to the excited hole ($q_A^h$ for atom A). For TDDFT, the sum of these charges is not guaranteed to equal 1, so we then normalize these charges:

$$q_{A,\text{norm}}^e = \frac{q_A^e}{\sum_A q_A^e} \qquad q_{A,\text{norm}}^h = \frac{q_A^h}{\sum_A q_A^h} \quad (4)$$

To calculate the IPR for the excited electron and hole for a given excitation, we then insert the normalized charges into the IPR equations:

$$\text{IPR}^e \equiv \frac{1}{N\sum_A \left(q_{A,\text{norm}}^e\right)^2} \qquad \text{IPR}^h \equiv \frac{1}{N\sum_A \left(q_{A,\text{norm}}^h\right)^2} \quad (5)$$

## Data availability

The optimized geometries for all of the quantum dots presented here are available in the standard xyz format in Supplementary Data 1. The TDDFT input files are included in Supplementary Data 2. All geometry optimizations were run using the commercially available QChem. The TDDFT calculations were run using a locally modified version of QChem, which allows for a larger number of CIS roots to be calculated. All other data generated and analyzed for this publication is available from the corresponding author upon request.

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

## Acknowledgements

This work was supported by the Department of Energy under grant number DMR-1905164. This work used the Extreme Science and Engineering Discovery Environment (XSEDE) resource Comet at the San Diego Supercomputing Center through allocation TG-DMR190054, which is supported by National Science Foundation grant number ACI-1548562. T.G. received financial support from the Technion-MIT fellowship and the Technion-New England Foundation. A.R.M. was supported by the National Science Foundation Graduate Research Fellowship under Grant No. 1122374.

## Author contributions

T.G. and A.R.M. contributed equally to this work. T.G. and A.R.M. performed all calculations and conducted all data analysis. T.V.V. conceived and oversaw the project. All authors contributed to writing the manuscript.

## Competing interests

The authors declare no competing interests.
