## [Peer Review File · Nature Communications]

REVIEWERS' COMMENTS

Reviewer #1 (Remarks to the Author):

The work from Goldzak et al. is a computational study on CdSe clusters which suggests that such systems are "inherently defective". Such finding is properly justified and rationalized in the manuscript, and in my opinion represents a noteworthy result. Such claim is grounded in the fact that even stoichiometric clusters with perfect passivation and without defects are characterized, in the low energy part of the spectrum, by dark transitions which are localized over the cluster surface.

The method employed to calculate the photoabsorption spectrum (TDDFT) is adequate, also basis set (LANL2DZ) as well as the exchange-correlation functional (PBE0) are adequate.

The paper is very well written, the References are representative (considering the wideness of the field!)

The topic of the work is quite important and can be interesting for a very wide scientific audience. Although there are previous CdSe photoabsorption calculations, it is the first time that this 'dark' transitions are studied in more detail. Moreover, also their importance is highlighted, which probably has been ignored or at least not properly considered in the previous works.

The authors have employed the proper tools (Density of Transitions, spectrum and IPR) in order to rationalize the results and to attribute the studied transitions to localized state over the surface.

In summary in my opinion the paper deserves publication in nature communications in its present form.

Reviewer #2 (Remarks to the Author):

The authors return to the long-standing problem of surface states in CdSe nanocrystals. They show that even with perfect passivation of dangling Cd bonds, which clears the forbidden gap from states, surface states owing to undercoordinated Se sites affect the absorption spectrum. They therefore argue for revisiting the importance of photochemistry and short-range photoinduced charge transfer to and from the nanocrystals.

These are important and original conclusions and I strongly recommend publication. I only have two minor points for the authors to consider.

1. In the abstract, the authors write: "As these chalcogenide atoms are not passivated by any commonly used ligand, one anticipates that varying the ligand will not dramatically change the number of dark states". I understand what the authors meant, but I think it would be good to change the phrasing such that it is clear that this refers to common, or Cd-related, ligands. Right now the term "varying the ligand" may be construed to mean any ligand whatsoever.

2. It may be helpful to mention early work from Chelikowsky's group [Troparevsky et al., JCP 119, 2284 (2003)], which is perhaps at the other extreme of the present work, as no ligands are used. And indeed a large "tail" in the absorption is also observed in that work.

Reviewer #1 (Remarks to the Author):

The work from Goldzak et al. is a computational study on CdSe clusters which suggests that such systems are "inherently defective". Such finding is properly justified and rationalized in the manuscript, and in my opinion represents a noteworthy result. Such claim is grounded in the fact that even stoichiometric clusters with perfect passivation and without defects are characterized, in the low energy part of the spectrum, by dark transitions which are localized over the cluster surface. The method employed to calculate the photoabsorption spectrum (TDDFT) is adequate, also basis set (LANL2DZ) as well as the exchange-correlation functional (PBE0) are adequate.

The paper is very well written, the References are representative (considering the wideness of the field!) The topic of the work is quite important and can be interesting for a very wide scientific audience. Although there are previous CdSe photoabsorption calculations, it is the first time that this 'dark' transitions are studied in more detail. Moreover, also their importance is highlighted, which probably has been ignored or at least not properly considered in the previous works.

The authors have employed the proper tools (Density of Transitions, spectrum and IPR) in order to rationalize the results and to attribute the studied transitions to localized state over the surface.

In summary in my opinion the paper deserves publication in nature communications in its present form.

We thank the reviewer for their time and their thoughtful review.

Reviewer #2 (Remarks to the Author):

The authors return to the long-standing problem of surface states in CdSe nanocrystals. They show that even with perfect passivation of dangling Cd bonds, which clears the forbidden gap from states, surface states owing to undercoordinated Se sites affect the absorption spectrum. They therefore argue for revisiting the importance of photochemistry and short-range photoinduced charge transfer to and from the nanocrystals.

These are important and original conclusions and I strongly recommend publication. I only have two minor points for the authors to consider.

1. In the abstract, the authors write: "As these chalcogenide atoms are not passivated by any commonly used ligand, one anticipates that varying the ligand will not dramatically change the number of dark states". I understand what the authors meant, but I think it would be good to change the phrasing such that it is clear that this refers to

common, or Cd-related, ligands. Right now the term "varying the ligand" may be construed to mean any ligand whatsoever.

Thank you for this suggestion, which we believe is very helpful and clarifies our point. We have modified this sentence to read:
"As chalcogenide atoms are not passivated by any Lewis base ligand..."

2. It may be helpful to mention early work from Chelikowsky's group [Troparevsky et al., JCP 119, 2284 (2003)], which is perhaps at the other extreme of the present work, as no ligands are used. And indeed a large "tail" in the absorption is also observed in that work.

Thank you for the suggestion, you are right that this paper is relevant as the "other extreme" of our work.

We have added a citation for this paper in the Introduction, for the sentence:
"A few studies have used *ab initio* tools to simulate optical absorption spectra of CdSe quantum dots"

And we have also added an acknowledgement of this work in the discussion:
"Looking at the small number of computational studies of the excited states of these NCs, TDDFT studies of Cd₃₃Se₃₃ with methylamine ligands had also found a tail in the absorption spectrum\cite{fischer_passivating_2012,del_ben_density_2011}, and studies using TDLDA for NCs with no ligands found an extremely broad and long tail\cite{troparevsky_optical_2003}, though these works did not explore the nature of these states."